# Wrinkle Formation and Initial Defect Sensitivity of Steered Tow in Automated Fiber Placement

**Meisam Kheradpisheh and Mehdi Hojjati \***

Department of Mechanical, Industrial, and Aerospace Engineering, Concordia University, 1455 De Maisonneuve Blvd W., Montreal, QC H3G 1M8, Canada; m_kherad@encs.concordia.ca
\* Correspondence: mehdi.hojjati@concordia.ca; Tel.: +1-514-848-2424 (ext. 5740)

**Abstract:** This paper aims to study the wrinkle formation of a prepreg with initial defect during steering in automated fiber placement (AFP). Wrinkle formation has a detrimental effect on the mechanical properties of the final product, limiting the AFP applications. A theoretical model for wrinkle formation has been developed in which a Pasternak foundation and a Koiter imperfection model are adapted to model viscoelastic characteristics of the prepreg tack and initial defect of the prepreg, respectively. The initial defect is defined as a slight deviation of the tow's mid-plane from a horizontal shape. The initial defect is generated in the tow by moving the tow through the guidance system, pressure of the roller, and resin tackiness. Galerkin method, along with the finite difference method (FDM), are employed to solve the wrinkle problem equation. The proposed method is able to satisfy the different boundary conditions for the wrinkle problem completely. The numerical results show that increasing the initial defect leads to a decrease in critical load and an increase in critical steering radius. To validate the theoretical model, experimental results are presented and compared with model-predicted results. It is shown that the model is well able to capture the trends and values of wrinkle formation wavelengths obtained from the experiment.

**Keywords:** wrinkle formation; prepreg; automated fiber placement; initial defect

## 1. Introduction

An increase in demand and operating conditions has resulted in the need for composite structures. To meet the increasing demand, automated fiber placement (AFP) offers excellent benefits that reduce costs and increase the production rate. Therefore, the AFP is becoming increasingly important for many applications in various industries. During the AFP process, the pre-impregnated tapes are placed on the tool, utilizing the AFP head. One of the significant advantages of AFP is that one manages to control the speed of each tow, and this advantage contributes to tow steering and the design and manufacturing of structures with complex geometries [1,2].

A primary concern during steering using automated fiber placement is the prepreg out-of-plane buckling, which significantly lowers the mechanical properties of the products [3]. Indeed, the steering curvature causes the inner edge of the tape to be under compression and the outer edge to be under tension. The values of compressive load and bending load applied to tow are controlled by the value of steering radius. A decrease in steering curvature is associated with increasing the compressive load. This compressive load causes out-of-plane wrinkle formation in the inner edge [3,4].

Although a considerable amount of literature has been published on buckling of laminate composite, the number of papers focusing on the buckling during the steering in automated fiber placement is not significant. Given the significance of control of different types of defects such as wrinkle formation in the AFP and manufacturing process, efforts have been made to study and understand these subjects. Panday and Sun [5] studied wrinkle formation of the composite laminate by two different methods. In the first approach, they modeled the interface bonding by a set of shear and normal springs and calculated the

buckling load for a composite laminate. The second approach employed a large deflection theory to obtain the governing equation for predicting wrinkle behavior. Ma et al. [6] studied the unilateral contact Local buckling of multilayered composite resting on elastic foundation. They employed the transfer function method to solve the buckling model and investigate the effect of elastic foundation parameters on buckling load. Beakou et al. [3] studied out-of-plate buckling of tape during the automated fiber placement process. They presented a simply supported buckling model of the plate under linearly varying in-plane load for the wrinkle formation of tow. Their experimental data showed that predicting the wrinkle formation can be improved by considering tack's temperature and dynamic behavior. Lightfoot et al. [7] investigated a new mechanism for the wrinkles formed in the prepreg composites due to thermal shear force. They showed that removing the release film decreases the possibility of wrinkle formation, which supported their proposed mechanism. The effect of steering curvature on wrinkle formation was performed by Matveev et al. [8]. They defined wrinkle formation as a buckling model for a plate resting on an elastic foundation and used experimental findings to predict the parameters of wrinkle formation. The effect of in-plane shear modulus on wrinkle formation was investigated [4]. They measured in-plane shear modulus of prepreg using the $\pm 45^\circ$ tensile test. Their results show that the shear modulus in the wrinkle equation leads to a better agreement with the experimental results. Bakhshi and Hojjati [9] performed a theoretical and experimental study on wrinkle formation of steered slit tow. The authors used the Rayleigh-Ritz and Laplace method to solve the problem of wrinkle formation. Besides, they presented a time-dependent elastic foundation model for modeling the tack properties. Rajan et al. [10] experimentally investigated the wrinkle formation in prepreg slit tape. They employed a Stereo DIC technology to measure the displacement and strains during and after experiments. They found the amplitude of wrinkles was related to the time and temperature and doubled after the placement process because of the time-dependent viscoelastic properties of the tape. Wehbe et al. [11] investigated the tow wrinkle on an arbitrary surface mathematically in the AFP process. They presented a mathematical model using the geodesic path and curvature definition that estimated the wrinkle amplitude. The effect of different parameters such as head speed and compaction force on steered tow on a cylindrical tool was studied by [12]. They evaluated the quality of the product based on the different combinations of these parameters and determined the importance of the parameters using the RReiliefF algorithm.

Imperfection in the composite laminate can be divided into two categories: imperfect interface and initial defect. In the former case, imperfection is defined as a weak bonding between layers [13,14]. This imperfection can be modeled as a thin interface joining two adjacent layers by distributed springs [15]. In the latter case, the initial defect is defined as a slight deviation of the midplane from the horizontal shape. During the AFP process, the initial defect may be caused in tow by the contact pressure of the roller and moving the tow through the guidance system. In recent years, researchers set out to study the effect of the initial defect (imperfection) in buckling problems. Shariat et al. [16] studied buckling load of simply supported functionally graded plates with an initial defect under uniform in-plane edge load. They employed the Galerkin method to simultaneously solve the buckling and compatibility equations to calculate the buckling load of the imperfect plate. Thermoelastic buckling of the imperfect orthotropic and isotropic plates under different thermal loading was investigated by [17]. Kiani et al. [18] presented an approximate close-form solution along with the Galerkin method to solve critical thermal buckling load for a sandwich FGM plate resting on Pasternak elastic foundation.

There is a growing need for detailed research on induced defects by automated fiber placement processes. As discussed above, there seems to be no detailed study on the wrinkle formation during the steering process considering the initial defect effect of the tape on the wrinkle. In this work, a theoretical model is presented for the wrinkle formation of the tape with an initial defect resting on the Pasternak foundation. Galerkin method in conjunction with the finite difference method are employed to analyze buckling load

and critical steering radius of tape and give an interpretation of how a slight initial defect affects these parameters. The method proposed in this study can be employed to solve wrinkle equations for different boundary conditions just by changing the finite difference coefficients. The model predictions for the values of buckling wavelengths are validated with experimental results.

## 2. Experiments

An AFP machine provided by Automated Dynamic Inc. was employed to fabricate the experimental samples on the surface, as shown in Figure 1a. This machine has a robot arm with six degrees of freedom, able to lay up both thermoset and thermoplastic composites. The number of tows used for adding lay-up by the head of the machine varies from one to four per each course. The tows can be cut in arbitrary lengths over the 3 inches and restarted during the lay-up process at any time. The AFP machine is equipped with a compaction roller and a hot gas as a heat source that provides pressure contact and a necessary temperature for bonding the tows on the tool, respectively. The compression force applied and controlled by the roller plays a vital role in the mechanical properties of tack and wrinkle formation. The compaction pressure distribution generated by roller on prepreg was investigated in [19,20], respectively. Besides, the compression force between the roller and tape and the guidance system leads to an initial defect in tape that affects the wrinkle formation. Figure 1b,c demonstrates the roller-tape contact and guiding system.

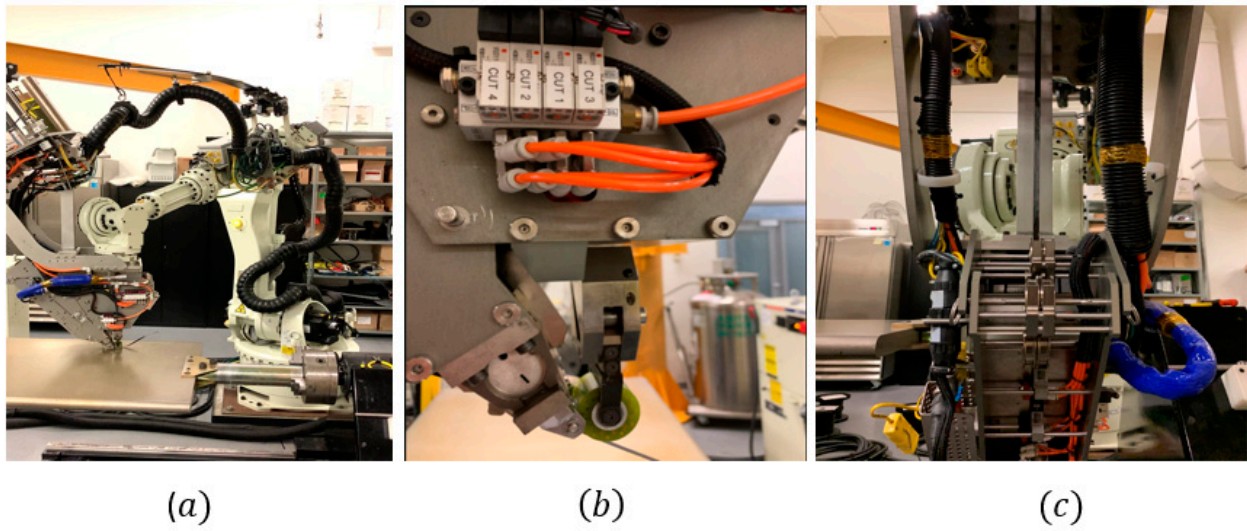

$(a)$ $(b)$ $(c)$

**Figure 1.** Automated fiber placement machine: (**a**) fiber placement with a six-axis robot arm, (**b**) the head and the roller-tape contact, (**c**) the guidance system.

The tows used in this experiment were carbon/epoxy unidirectional prepreg (CYCOM 977–2/HTS-145) from Solvay with an individual tow width of 6.35 mm, thickness of 0.2 mm, and a 60% fiber volume. The prepregs were made of 12K carbon fibers (HTS-145), which were pre-impregnated with CYCOM 977-2 epoxy resin system and cured using an autoclave or press molding.

To get experimental results, firstly, the thermoset prepregs were left to rest for 30 min at room temperature. A sketch was made by AFP machine's software for determining the tow's path on an aluminum tool. The dimensions of this path are shown in Figure 2.

Then, the thermoset prepregs were left to rest for 30 min at room temperature. After that, the tows were stored on the lay-up head and directly delivered from the spools to the roller to be steered on the rigid aluminum tool. The aluminum tool was cleaned with acetone before each experiment to reduce the shear force created by friction between the tool and prepreg. Finally, prepreg tows were deposited on an aluminum tool with different steering radii of 55, 60, 65, 67, 70, and 75 cm, according to Figure 2. The process of steering

the tape was proceeded for this experiment by adjusting the AFP software according to Table 1.

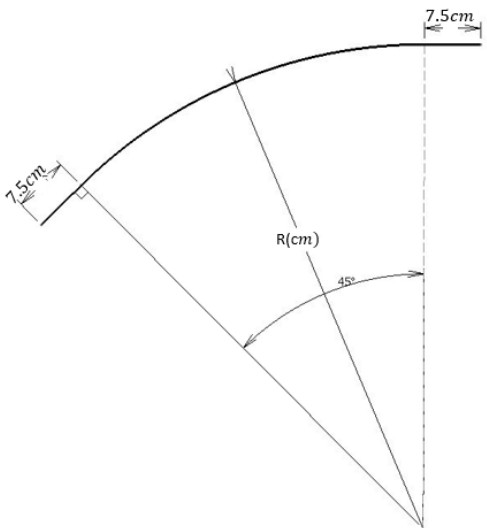

**Figure 2.** The tow's path made by AFP machine's software.

**Table 1.** Process conditions.

| Head Speed | Hot Gas Temperature (Heat Source) | Flow Rate (Heat Source) | Roller Force | Tool Temperature |
|---|---|---|---|---|
| 77 mm/s | 160 °C | 85 (L/min) | 250 N | 23 °C |

The steering radius is created by rotating the head. The head rotation generates a linearly varying load in prepreg, which in turn can lead to tow buckling.

The experimental work observed that the wrinkle is formed when the tapes are placed on the surface with a radius of less than a specific value (critical radius). Figure 3 demonstrates the steered tow and wrinkle formation (out-of-plane buckling) in the tow.

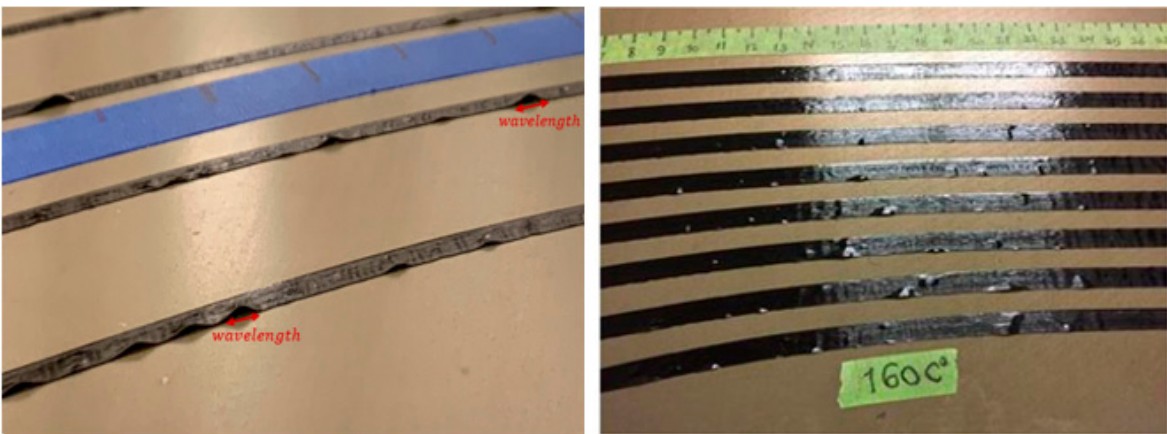

**Figure 3.** Steered prepreg on aluminum tool and wrinkle formation generated in tows.

One of the critical parameters in the process of steering the tows is the wavelength of the wrinkle, defined as the length of an opening separation of an individual wrinkle (see Figures 3 and 4).

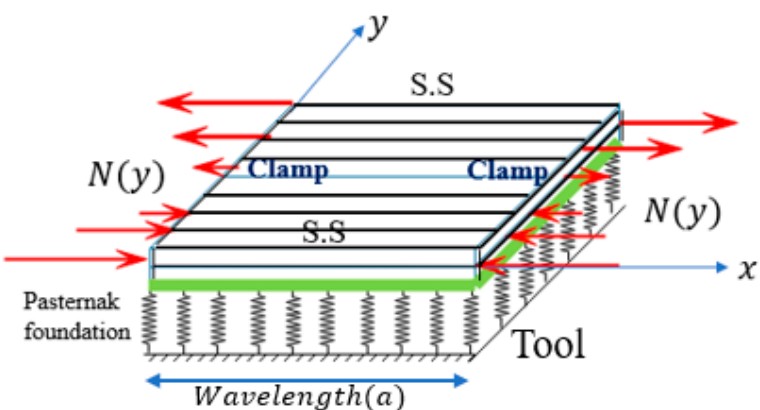

**Figure 4.** Schematic of a section of tape resting on elastic foundation under combined axial compression and bending.

The length of the wrinkles (wavelength) is measured using a caliper manually in each tow. The average of the wrinkle's lengths in each steered tow is considered the wrinkles' wavelength in that certain radius. Table 2 represents the number of wavelengths and the average length of wrinkles in each steering radius.

**Table 2.** Measurement results (the number of wrinkles and the average length of wrinkles in each radius).

| Steering Radius (cm) | Number of Wrinkles | Average of Wavelength (mm) |
|---|---|---|
| 55 | 16 | 6.2 |
| 60 | 15 | 6.6 |
| 65 | 14 | 7 |
| 67 | 14 | 7.4 |
| 70 | 9 | 8 |
| 75 | 6 | 9 |

In the following, based on the experiments, the authors aim to present a developed buckling formulation to model the wrinkle formation in tows. To validate the proposed model, the wavelengths of wrinkles measured manually in experiments are compared with the model predictions.

## 3. Formulation

### 3.1. Buckling Model

Out-of-plane wrinkle formation of steered tows during the automated fiber placement process can be modeled as a buckling problem of a rectangular plate resting on a Pasternak elastic foundation, which is under non-uniform load. The load is applied by a roller to two clamped edges, and the other two edges are considered as a simply support (S.S) condition in accordance with the geometry of the problem. Figure 4 shows the theoretical model of wrinkle formation during steering.

### 3.1.1. Pasternak Elastic Foundation

In this paper, a Pasternak foundation representation is adopted to model the mechanical properties of the tool and prepreg tack [21,22]. Figure 5 shows the free body diagram of adhesive joint and elastic modeling based on the Pasternak model. This model replaces the elastic foundation as a combination of the shear layer and normal linear spring. As a result, the pressure of the elastic foundation surface can be mathematically written as:

$$p = K_f w - G \nabla^2 w \tag{1}$$

where $p$ is pressure, $\nabla^2$ is Laplace operator, and $K_f$ and $G$ are spring and shear constants of the Pasternak model, respectively.

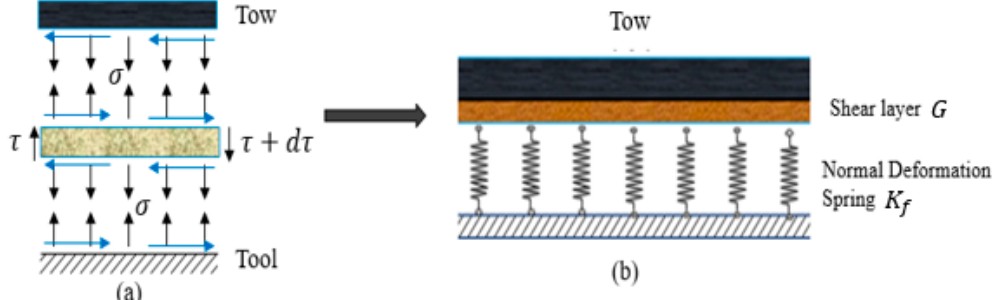

**Figure 5.** Elastic foundation: (**a**) free body diagram adhesive joint, (**b**) Pasternak elastic foundation model.

### 3.1.2. Governing Equation

Assuming that the thickness is constant through the z-direction and using classical lamination plate theory CLPT theory, the equilibrium equations for a general thin composite plate can be expressed as [23]:

$$\frac{\partial N_{xx}}{\partial x} + \frac{\partial N_{xy}}{\partial y} = 0$$
$$\frac{\partial N_{xy}}{\partial x} + \frac{\partial N_{yy}}{\partial y} = 0$$
$$\frac{\partial^2 M_{xx}}{\partial x^2} + 2\frac{\partial^2 M_{xx}}{\partial x \partial y} + \frac{\partial^2 M_{yy}}{\partial y^2} + N_{xx}\frac{\partial^2 w}{\partial x^2} + 2N_{xy}\frac{\partial^2 w}{\partial x \partial y} + N_{yy}\frac{\partial^2 w}{\partial y^2} + p = 0$$

(2)

where $w$ is the displacement component along z direction; $p$ is the elastic foundation pressure; and $(N_{xx}, N_{xy}, N_{yy})$ and $(M_{xx}, M_{xy}, M_{yy})$ are the in-plane force and moment resultants, respectively, that can be defined as:

$$\begin{bmatrix} N_x \\ N_y \\ N_{xy} \end{bmatrix} = \begin{bmatrix} A_{11} & A_{12} & A_{16} \\ A_{12} & A_{22} & A_{26} \\ A_{16} & A_{26} & A_{66} \end{bmatrix} \begin{bmatrix} \varepsilon_x^0 \\ \varepsilon_y^0 \\ \varepsilon_{xy}^0 \end{bmatrix} + \begin{bmatrix} B_{11} & B_{12} & B_{16} \\ B_{21} & B_{22} & B_{26} \\ B_{16} & B_{26} & B_{66} \end{bmatrix} \begin{bmatrix} \kappa_x^0 \\ \kappa_y^0 \\ \kappa_{xy}^0 \end{bmatrix}$$
$$\begin{bmatrix} M_{xx} \\ M_{yy} \\ M_{xy} \end{bmatrix} = \begin{bmatrix} B_{11} & B_{12} & B_{16} \\ B_{21} & B_{22} & B_{26} \\ B_{16} & B_{26} & B_{66} \end{bmatrix} \begin{bmatrix} \varepsilon_x^0 \\ \varepsilon_y^0 \\ \varepsilon_{xy}^0 \end{bmatrix} + \begin{bmatrix} D_{11} & D_{12} & D_{16} \\ D_{21} & D_{22} & D_{26} \\ D_{16} & D_{26} & D_{66} \end{bmatrix} \begin{bmatrix} \kappa_x^0 \\ \kappa_y^0 \\ \kappa_{xy}^0 \end{bmatrix}$$

(3)

where $A_{ij}$, $B_{ij}$, and $D_{ij}$ are the extensional stiffness coefficients, the coupling stiffness coefficients, and the bending stiffness coefficients, respectively. These matrices are obtained from Equation (4):

$$(A_{ij}, B_{ij}, D_{ij}) = \int_{-\frac{t}{2}}^{\frac{t}{2}} [\overline{Q}_{ij}] \left(1, z_i, z_i^2\right) dz_i$$

(4)

where $\overline{Q}_{ij}$ is transformed reduced stiffnesses, and $z_i$ and t are the coordinates in the z-direction and the thickness of the tow, respectively. Since the prepreg is an orthotropic material, the matrix $[B]$ is zero, and $D_{16}, D_{26} = 0$. By direct substitution of the moment and results from Equation (3) into Equation (2), the governing equation of a unidirectional prepreg tape resting on the Pasternak elastic foundation can be obtained by the following equation [23,24].

$$\left[D_{11}\frac{\partial^4 w}{\partial x^4} + 2(D_{12} + 2D_{66})\frac{\partial^4 w}{\partial x^2 \partial y^2} + D_{22}\frac{\partial^4 w}{\partial y^4}\right] - N_x\left(\frac{\partial^2 w}{\partial x^2}\right) + K_f w -$$
$$G\left(\frac{\partial^2 w}{\partial x^2} + \frac{\partial^2 w}{\partial y^2}\right) = 0$$

(5)

where $k$, $G$ are spring constants of Pasternak model, and according to the physics of the problem, $N_x$ is a linearly varying in-plane load that can be defined by:

$$N_x = -N\left(1 - \frac{\eta y}{b}\right)$$

(6)

where $b$ is the width of the tow, and $\eta$ is a constant between 0 and 2, for example, $\eta = 0$ and $\eta = 2$ correspond to uniform compressive load and pure bending load, respectively. Figure 6 demonstrates the in-plane load distribution for various values of $\eta$.

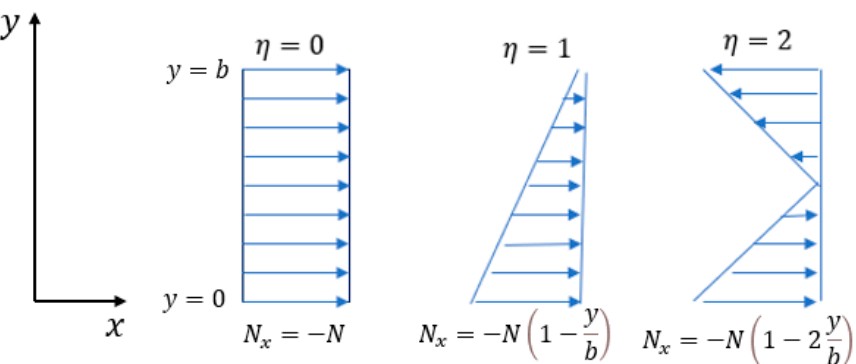

**Figure 6.** The stress distribution for different values of $\eta$.

### 3.2. Initial Defect

There are two types of imperfection that might happen, which include the interfacial imperfection and the initial geometric defect (imperfection). In this problem, the out-of-plane buckling is investigated for AFP. Therefore, the initial geometric defect, which means a slight deviation of the middle plane of prepreg from a flat shape, is modeled by Equation (7). The interface imperfection is modeled by the Pasternak elastic foundation equations (see Equation (1)). In this paper, $w^*(x,y)$ represents the initial defect. In this regard, the slight initial defect is defined based on the Koiter model as [25,26].

$$w^*(x,y) = \mu h w \tag{7}$$

where $\mu h$ represents the amplitude of the initial defect (imperfection), and $(0 \leq \mu < 1)$ when the $\mu = 0$, it expresses a perfect lamina. To consider the initial defects, the vertical displacement $w$ in Equation (5) needs to be replaced by $w + w^*$. It should be noticed that the expression in the bracket in Equation (5) remains constant for the reason that this expression resulted from the bending moments. The bending moments are only related to the curvature of the plate, and they remain unchanged by changing the total curvature. As a result, Equation (5) yields:

$$\left[ D_{11} \frac{\partial^4 w}{\partial^4 x} + 2(D_{12} + 2D_{66}) \frac{\partial^4 w}{\partial^2 x \partial^2 y} + D_{22} \frac{\partial^4 w}{\partial^4 y} \right] - N_x \left( w_{,xx} + w^*_{,xx} \right) + K_f (w + w^*) - G \left( w_{,xx} + w^*_{,xx} + w_{,yy} + w^*_{,yy} \right) = 0 \tag{8}$$

To obtain the values of wavelength or critical force of wrinkle formation, one needs to solve Equation (8).

### 3.3. Solution Procedure

To solve the PDE Equation (8), since the edges loaded are clamped, the solution can be presented in the form:

$$w(x,y) = w_m(y) \left( 1 - \cos\left( 2 \frac{m\pi x}{a} \right) \right) \tag{9}$$

where $\left( 1 - \cos\left( 2 \frac{m\pi x}{a} \right) \right)$ satisfies the boundary condition in $x = 0, a$. By definition of initial defects based on the Koiter model, the $w^*(x,y)$ can be expressed as:

$$w^*(x,y) = \mu h w_m(y) \left( 1 - \cos\left( 2 \frac{m\pi x}{a} \right) \right) \tag{10}$$

Substitution of $w(x,y)$ and $w^*(x,y)$ into Equation (8), and after rearrangements, one can deduce the following equation:

$$
\begin{aligned}
R_1 = {} & D_{22}\left(1 - \cos\left(2\tfrac{m\pi x}{a}\right)\right)\frac{d^4 w_m(y)}{dy^4} \\
& + 2(D_{12} + 2D_{66})\frac{4\pi^2 m^2}{a^2}\frac{d^2 w_m(y)}{dy^2}\cos\left(2\tfrac{m\pi x}{a}\right) - G(1 \\
& + \mu h)\frac{d^2 w_m(y)}{dy^2}\left(1 - \cos\left(2\tfrac{m\pi x}{a}\right)\right) \\
& - D_{11}\frac{16\pi^4 m^4}{a^4}\cos\left(2\tfrac{m\pi x}{a}\right)w_m(y) \\
& - N_x\left(\frac{4\pi^2 m^2}{a^2}(1 + \mu h)\right)w_m(y)\cos\left(2\tfrac{m\pi x}{a}\right) \\
& + K_f(1 + \mu h)w_m(y)\left(1 - \cos\left(2\tfrac{m\pi x}{a}\right)\right) \\
& - G\left(\frac{4\pi^2 m^2}{a^2}(1 + \mu h)\right)\cos\left(2\tfrac{m\pi x}{a}\right)w_m(y)
\end{aligned}
\tag{11}
$$

By direct substitution of $N_x = -N\left(1 - \frac{\eta y}{b}\right)$ into Equation (11), it leads to a non-linear and non-homogeneous ordinary differential equation. To find the solution for Equation (11), first we transform it from a non-homogenous to a homogenous one using the Galerkin method. To reach a homogenous equation, Equation (11) should be multiplied by an admissible function satisfying the boundary condition. The following function has the conditions of the Galerkin method.

$$
\psi = \left(1 - \cos\left(2\frac{n\pi x}{a}\right)\right)
\tag{12}
$$

According to the Galerkin method:

$$
\int_0^a R_1 \times \psi \, dx = 0
\tag{13}
$$

In the calculation of the above integrals, two integrals appear, which can be solved as follows:

$$
\int_0^a \left(1 - \cos\left(2\frac{m\pi x}{a}\right)\right) \times \left(1 - \cos\left(2\frac{n\pi x}{a}\right)\right) dx = \begin{cases} \frac{3}{2}a & m = n \\ a & m \neq n \end{cases}
\tag{14}
$$

$$
\int_0^a \left(\cos\left(2\frac{m\pi x}{a}\right)\right) \times \left(1 - \cos\left(2\frac{n\pi x}{a}\right)\right) = \begin{cases} -\frac{1}{2}a & m = n \\ 0 & m \neq n \end{cases}
\tag{15}
$$

By employing the Galerkin method, the non-homogenous Equation (11) transform the homogenous one, which after rearrangements, can be written as:

$$
\begin{aligned}
& D_{22}\frac{d^4 w_m(y)}{dy^4} - \left[2(D_{12} + 2D_{66})\frac{4\pi^2 m^2}{3a^2} + G(1 + \mu h)\right]\frac{d^2 w_m(y)}{dy^2} \\
& + \left[D_{11}\frac{16\pi^4 m^4}{3a^4} + K_f(1 + \mu h) + G\left(\frac{4\pi^2 m^2}{3a^2}(1 + \mu h)\right)\right]w_m(y) \\
& - N\left(1 - \frac{\eta y}{b}\right)\left(\frac{4\pi^2 m^2}{3a^2}(1 + \mu h)\right)w_m(y) = 0
\end{aligned}
\tag{16}
$$

Now, as mentioned in the problem definition, the boundary conditions of unloaded edges can be considered S.S according to the physics of the problem. The boundary condition equations for S.S are presented as:

$$
\begin{aligned}
w & = 0 \\
M_{,yy} & = D_{12}\frac{d^2 w}{dx^2} + D_{22}\frac{d^2 w}{dy^2} = 0
\end{aligned}
\tag{17}
$$

To write the boundary conditions in the form of $w(y)$, we substitute $w$ from Equation (9) into Equation (17); the simply supported (S.S) equations result in:

$$
\begin{aligned}
w & = 0 \\
M_{,yy} & = D_{12}\frac{4\pi^2 m^2}{a^2}w(y)\cos\left(2\tfrac{m\pi x}{a}\right) + D_{22}\frac{d^2 w}{dy^2}\left(1 - \cos\left(2\tfrac{m\pi x}{a}\right)\right) \approx 0
\end{aligned}
\tag{18}
$$

Applying the Galerkin method to the Equation (18), the boundary conditions lead to:

$$w = 0$$
$$M_{,yy} = D_{22}\frac{d^2w(y)}{dy^2} - \frac{4\pi^2 m^2}{3a^2}D_{12}w(y) = 0 \tag{19}$$

To solve the nonlinear homogenous ordinary Equation (16), The finite difference method (FDM) was employed [27]. In this method, first, the interval between $y = 0$ and $y = b$ is partitioned into $n$ subintervals. Then, the differential operators are approximated by the differential quotients. Based on the FD method, differential operators can be expressed in the following forms:

$$\begin{aligned}
\left(\frac{dw}{dy}\right)_i &= \frac{1}{2s}(-w_{i-1} + w_{i+1}) \\
\left(\frac{d^2w}{dy^2}\right)_i &= \frac{1}{s^2}(w_{i-1} - 2w_i + w_{i+1}) \\
\left(\frac{d^3w}{dy^3}\right)_i &= \frac{1}{2s^3}(-w_{i-2} + 2w_{i-1} - 2w_{i+1} + w_{i+2}) \\
\left(\frac{d^4w}{dy^4}\right)_i &= \frac{1}{s^4}(w_{i-2} - 4w_{i-1} + 6w_i - 4w_{i+1} + w_{i+2})
\end{aligned} \tag{20}$$

where $i$ is the node number, $w_i$ represents the deflection component in z-direction for the $i$th node, and $s = b/n$ is the distance between the two nodes. Figure 7 shows the sections in the y-direction. According to Figure 7, the coordinates of the nodes in the y-direction can be written as:

$$y_i = s(i - 1) \tag{21}$$

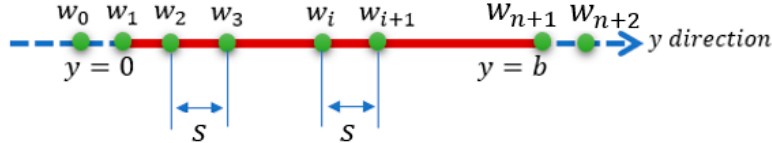

**Figure 7.** Staggered grid and nodes in the y-direction.

By replacing Equations (20) and (21) into Equation (16), and after rearrangements, the nonlinear equation can be written as:

$$X_1 w_i - X_2 Z_i + X_3 U_i - X_4 F_i w_i = 0 \tag{22}$$

where

$$\begin{aligned}
X_1 &= \left[D_{11}\frac{16\pi^4 m^4}{3a^4} + K_f(1 + \mu h) + G\frac{4\pi^2 m^2}{3a^2}(1 + \mu h)\right] \\
X_2 &= \left[G(1 + \mu h) + 2(D_{12} + 2D_{66})\frac{4\pi^2 m^2}{3a^2}\right]\frac{1}{s^2} \\
X_3 &= D_{22}\frac{1}{s^4} \\
X_4 &= N\frac{4\pi^2 m^2}{3a^2}(1 + \mu h) \\
Z_i &= (w_{i-1} - 2w_i + w_{i+1}) \\
U_i &= w_{i-2} - 4w_{i-1} + 6w_i - 4w_{i+1} + w_{i+2} \\
F_i &= \left(1 - \frac{\eta(i-1)}{n}\right)
\end{aligned} \tag{23}$$

As it can be seen in Figure 7, there are two nodes named $w_0, w_{n+2}$ which are out of the trivial difference between $y = 0$ and $y = b$. Theses nodes are defined by boundary conditions at $y = 0, b$. Replacing $w_1$ into the boundary condition equations at $y = 0$ leads to:

$$w = 0 \Rightarrow w_1 = 0$$
$$D_{22}\frac{d^2w(y)}{dy^2} - \frac{4\pi^2 m^2}{3a^2}D_{12}w(y) = 0 \Rightarrow -D_{12}\frac{4\pi^2 m^2}{3a^2}w_1 + D_{22}\frac{1}{s^2}(w_0 - 2w_1 + w_2) = 0 \xrightarrow{w_1=0} w_0 = -w_2 \tag{24}$$

Again, the substitution of $w_{n+1}$ into the boundary condition equation at $y = b$ leads to:

$$w = 0 \Rightarrow w_{n+1} = 0$$
$$D_{22}\frac{d^2 w(y)}{dy^2} - \frac{4\pi^2 m^2}{3a^2}D_{12}w(y) = 0 \Rightarrow -D_{12}\frac{4\pi^2 m^2}{3a^2}w_1 + D_{22}\frac{1}{s^2}(w_n - 2w_{n+1} + w_{n+2}) = 0 \overset{w_{n+1}=0}{\Rightarrow} w_{n+2} = -w_n \tag{25}$$

With regard to the simply supported conditions, $w_{n+1}$, and $w_1$ are zero. Therefore, Equation (22) for each node can be obtained. By substitution of $w_i$ for $(i = 2, \ldots, n)$ into Equation (22), we have a linear system of $n-1$ equations for the $n-1$ unknowns $[w_2,\ w_3,\ \ldots,\ w_n]$. This equation system can be written in matrix form:

$$\left\{ X_1 \begin{bmatrix} 1 & 0 & \cdots & 0 \\ 0 & 1 & \cdots & 0 \\ 0 & 0 & \cdots & 0 \\ \vdots & \vdots & \ddots & \vdots \\ 0 & 0 & \cdots & 0 \\ 0 & 0 & \cdots & 1 \end{bmatrix}_{n-1 \times n-1} - X_2 \begin{bmatrix} -2 & 1 & 0 & 0 & \vdots & 0 & 0 & 0 \\ 1 & -2 & 1 & 0 & \vdots & 0 & 0 & 0 \\ 0 & 1 & -2 & 1 & \vdots & 0 & 0 & 0 \\ \vdots & \vdots & \vdots & \vdots & \vdots & \vdots & \vdots & \vdots \\ 0 & 0 & 0 & 0 & \vdots & 1 & -2 & 1 \\ 0 & 0 & 0 & 0 & \vdots & 0 & 1 & -2 \end{bmatrix}_{n-1 \times n-1} \right.$$

$$+ X_3 \begin{bmatrix} 5 & -4 & 1 & 0 & 0 & \vdots & 0 & 0 & 0 & 0 & 0 \\ -4 & 6 & -4 & 1 & 0 & \vdots & 0 & 0 & 0 & 0 & 0 \\ 1 & -4 & 6 & -4 & 1 & \vdots & 0 & -4 & -4 & 0 & 0 \\ \vdots & \vdots & \vdots & \vdots & \vdots & \vdots & \vdots & \vdots & \vdots & \vdots \\ 0 & 0 & 0 & 0 & 0 & \vdots & 0 & 1 & -4 & 6 & -4 \\ 0 & 0 & 0 & 0 & 0 & \vdots & 0 & 0 & 1 & -4 & 5 \end{bmatrix}_{n-1 \times n-1}$$

$$\left. - X_4 \begin{bmatrix} 1 & 0 & \cdots & 0 \\ 0 & 1-\eta/n & \cdots & 0 \\ 0 & 0 & \cdots & 0 \\ \vdots & \vdots & \ddots & \vdots \\ 0 & 0 & \cdots & 0 \\ 0 & 0 & \cdots & 1-\eta(n-1)/n \end{bmatrix}_{n-1 \times n-1} \right\} \begin{bmatrix} w_2 \\ w_3 \\ w_4 \\ \vdots \\ w_n \end{bmatrix} = \tag{26}$$

To have a non-trivial solution for the above system of equations, the determinant of the matrix expression in the bracket is to be equal to zero. This determinant is an expression for $a$ (wrinkle wavelength) and $N$ (critical buckling load).

### 3.4. Relation between Critical Steering Radius and Critical Load

Owing to the unbalanced length during the steering process, a linearly varying in-plane load is generated in tow.

This load is shown in Figure 4. To find a relation between steering radius and the applied load, the bending moment and curvature equation can be employed. Figure 8 shows the tape under a bending load. The bending moment and curvature are linked to each other by the following equation.

$$R = \frac{E_1 I}{M_o} \tag{27}$$

where $E_1$ is Young's modulus in the fiber direction, $I$ is the moment of inertia for the tow, $R$ is the steering radius, and $M_o$ is the memont about point O (see Figure 8).

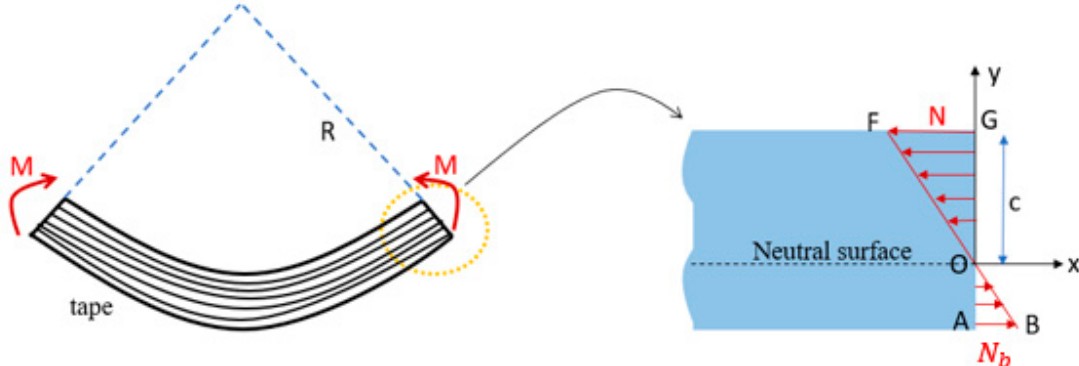

**Figure 8.** A schematic diagram of a steered tow under the bending load generated by the roller.

Regarding Figure 8, the c is equal to $b/\eta$ and from the similarity of two triangles $\Delta AOB$ and $\Delta FOG$, the value of $N_b$ (the tensile force at the outer edge) can be found:

$$\Delta FTO \sim \Delta AOB \Rightarrow \frac{N}{N_b} = \frac{\frac{b}{\eta}}{b\left(1 - \frac{1}{\eta}\right)} \Rightarrow N_b = N(\eta - 1) \tag{28}$$

As a result, the value of moment about the natural axis (point O) yields:

$$M_o = N\frac{b^2}{3\eta^2} + N\frac{b^2(\eta - 1)^3}{3\eta^2} \tag{29}$$

The moment of inertia of the cross-section with respect to the x-axis is obtained from Equation (30)

$$I = tb^3\left(\frac{1}{12} + \left(\frac{1}{2} - \frac{1}{\eta}\right)^2\right) \tag{30}$$

where $t$ and $b$ represent the thickness and width of the tape, respectively. Substituting Equations (29) and (30) obtained for M and I into Equation (27), one finds that:

$$R = \frac{E_1 I}{M_o} = \frac{E_1 tb\left(\frac{1}{12} + \left(\frac{1}{2} - \frac{1}{\eta}\right)^2\right)}{N\left(\frac{1}{3\eta^2} + \frac{(\eta - 1)^3}{3\eta^2}\right)} \tag{31}$$

Thus, if the maximum induced load ($N$) obtained from Equation (31) for the various values of the steering radius is more than the buckling load obtained from Equation (26), the wrinkle will occur in tow. In contrast, the tows will be wrinkle-free if the induced load is less than the buckling load (Equation (26)).

## 4. Result and Discussion

### 4.1. Numerical Results Obtained from Wrinkle Model

For the calculations, the mechanical properties of prepreg used in this paper were measured in a previous paper published by Bakhshi and Hojjati [9]. Table 3 indicates these mechanical properties.

**Table 3.** Material properties of prepreg and elastic foundation [9].

| $E_1$ (Gpa) | $E_2$ (Mpa) | $G_{12}$ (Mpa) | $\eta$ | $v_{12}$ | $t$ (mm) | $b$ (mm) |
|---|---|---|---|---|---|---|
| 31 | 0.046 | 3.025 | 2 | 0.2 | 0.2 | 6.35 |

Another important point to be mentioned is that since the wrinkle forms in tow as the first buckling load, the value of m in Equations (9) and (10) should be one.

### 4.1.1. The Effect of Initial Defect on Critical Load and Steering Radius

As mentioned before, the initial geometric defect (imperfection) is introduced as a slight deviation of the midplane from a flat one. The impact of increasing the initial defect on buckling load and steering radius is presented in Figures 9 and 10. The values of spring constants assume to be k = $3.25 \times 10^8$ (N/m) and G = 605 N/m according to [4,9] in the mathematical wrinkle model. As can be seen, a rise in initial defect is associated with a small reduction in critical load, which, according to Equation (31), implies an increase in minimum steering radius. In Figures 9 and 10, the value of *h* is assumed to be one.

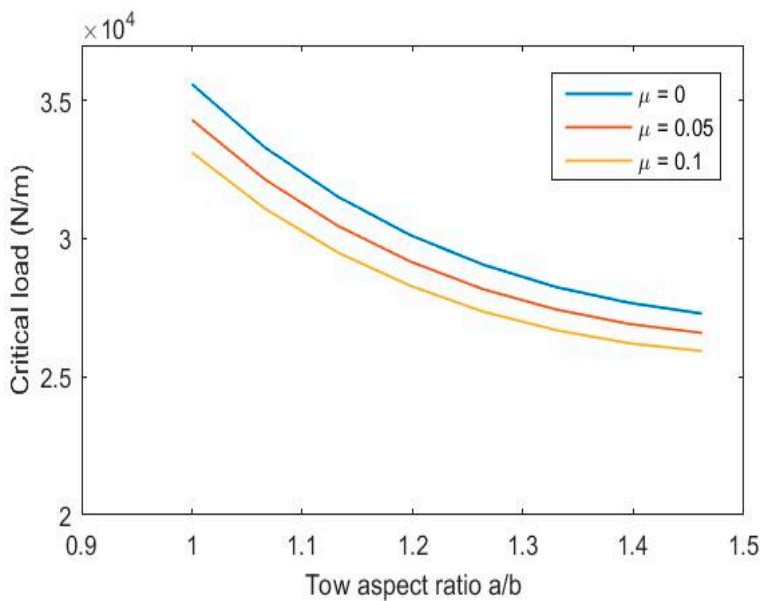

**Figure 9.** The variations of critical load versus different plate aspect ratios for various initial defect coefficients.

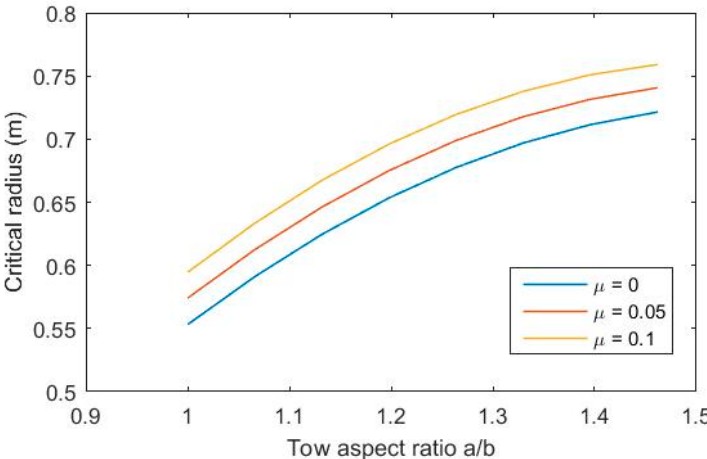

**Figure 10.** The variations of critical radius versus different plate aspect ratios and initial defect coefficients.

### 4.1.2. The Effects of Tack Stiffness (Spring Constant of the Pasternak Model) and Initial Defect (Imperfection) of the Tape on the Minimum Steering Radius

The variations of steering radius and buckling load with stiffness parameter of Pasternak foundation model for a tow with different initial defects are displayed in

Figures 11 and 12, respectively. The most important observations are as follows. As the stiffness parameter increases, the critical radius declines, and the buckling load increases. This can be explained by the fact that increasing the stiffness ($K_f$) means a stronger tack. Thus, to overcome the tack of prepreg, the axial in-plane load applied to the prepreg plate should be increased.

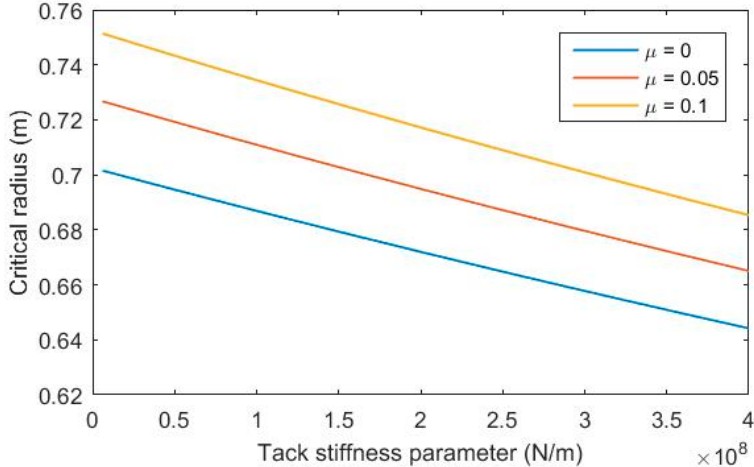

**Figure 11.** Critical radius steering variations with respect to tack stiffness parameter for different initial defect coefficients.

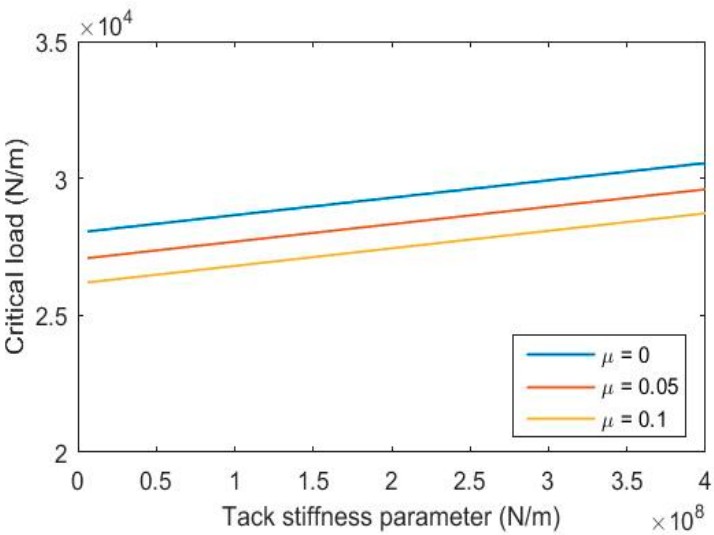

**Figure 12.** Critical buckling load variations with respect to tack stiffness parameter for different initial defect coefficients.

In contrast, according to Equation (30), the steering radius has an inverse relation with the buckling load. As a result, an increasing in-plane load leads to a decrease in radius steering. Besides, a rise in the value of the initial defect of the tow based on the Koiter model leads to a slight growth in critical steering radius and a slight reduction in buckling load. In Figures 11 and 12, the value of the aspect ratio for the tow is assumed to be 1.5.

*4.2. Drawing a Comparison between Wrinkle Wavelengths Obtained from the Theoretical Model and Experimental Work*

To validate the theoretical model with experiment results, in this section, a comparison is made between the values of the wrinkle wavelengths calculated from the theoretical wrinkle model and the experimental findings. For this reason, the value of maximum load from Equation (30) as a function of critical radius is replaced into Equation (25) instead

of $N$. By solving the determinant of Equation (25) for wrinkle wavelength ($L$), the value of wavelength is found for each critical radius. Figure 13 shows steering radius versus wrinkle wavelength for both experimental and theoretical results. The values of spring constants assume to be k = 3.25 × 10$^8$ (N/m) and G = 605 N/m according to [4,9] in the mathematical wrinkle model.

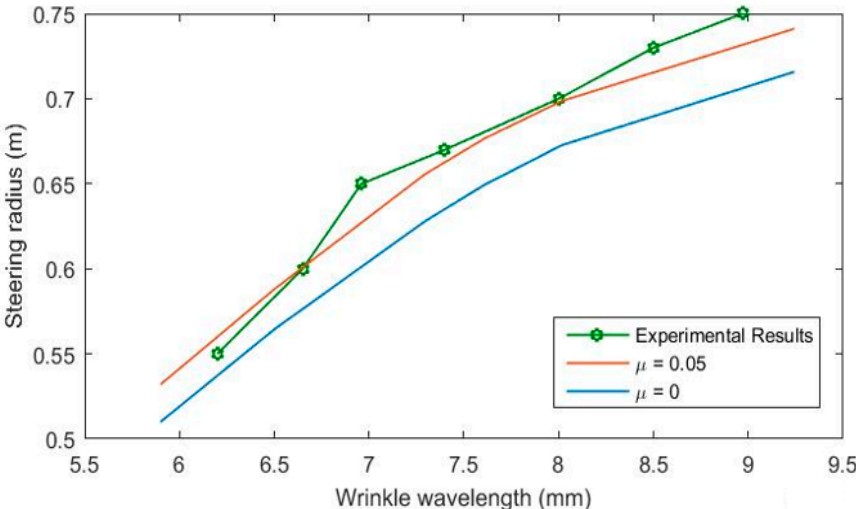

**Figure 13.** Wrinkle wavelength of the tape versus different steering radius.

As can be seen, the results yielded from the proposed model are in good agreement with experiment results. Also, it is found that considering the effect of a slight defect (imperfection) with an initial defect coefficient of $\mu = 0.05$ leads to a better agreement between the proposed model and experiment results.

## 5. Conclusions

This study is concerned with the wrinkle formation during steering with an initial defect based on the Koiter model. A theoretical buckling model is presented to model the wrinkle formation of the prepreg tape resting on an elastic foundation. Pasternak model is adopted to model tack properties. The Galerkin method along with the finite difference method are employed to solve the problem for critical buckling load and steering radius. The solution method presented in this study can completely satisfy the different boundary conditions of the problem. The obtained results reveal that an increasing initial defect is associated with an overall reduction in buckling load, and consequently, the smaller critical loads occur for higher critical radius steering values. Any improvement in tack (elastic foundation) properties significantly affects the numerical values of the critical load and radius. Specifically, growth in foundation stiffness results in a notable reduction in the critical radius, which subsequently leads to increasing the load. Finally, the results for wrinkle wavelengths yielded from the applied model are compared to those of experimental work. It is worth noting that the diagrams showed that the model predictions are very close to the results measured from experimental work.

**Author Contributions:** M.K.: Methodology, Formal analysis, Software, Writing-Original Draft. M.H.: Conceptualization, Validation, Writing-Review Editing, Supervision, Project administration, Funding acquisition. All authors have read and agreed to the published version of the manuscript.

**Funding:** This research received no external funding.

**Acknowledgments:** The authors gratefully acknowledge financial support from the Natural Sciences and Engineering Research Council of Canada (NSERC). We also would like to thank Bombardier Aerospace for providing the towpreg materials.

**Conflicts of Interest:** We declare that this research is original, has not been published elsewhere, and is not currently being considered for publication elsewhere. We know of no conflicts of interest associated with this publication, and there has been no significant financial support for this work that could have influenced its outcome. As Corresponding Author, I confirm that the manuscript has been read and approved for submission by all the named authors.

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
