# Peer review of "Wrinkle Formation and Initial Defect Sensitivity of Steered Tow in Automated Fiber Placement"

_jcs, doi:10.3390/jcs5110295_

Round 1

Reviewer 1 Report

The authors investigate an interesting and complex problem of defect formation during fiber steering.

The manuscript needs major revision for the following reasons:

  • missing reference for the claim on line 36-37
  • the experimental procedure to measure wrinkle wavelength (WL) and critical radius (CR) are not presented. Furthermore,  WL is not defined
  • there is a significant difference between the the experimental critical radiuses that ranges from 0.5 to 0.75 m (fig 12) and that obtained from the theoretical model that spans from 20 to 120 m (fig. 9)
  • tack stiffness is not defined

Other issues:

  - in table 1 instead of lit/s is SLPM (standard liter per minute)

 -  line 141, the head is rotating while the roller is fixed. In my opinion The roller acts as a clamp while the head rotates and generates  the observed buckling

- imperfections should be replace by defects

Reviewer 2 Report

Please refer to the attached review report regarding comments and suggestions.

Round 2

Reviewer 1 Report

The authors addressed all the reviewer's comments, therefore the paper is ready to be published

Reviewer 2 Report

The revisions are satisfactory.